# Global and national trends, gaps, and opportunities in documenting and monitoring species distributions

**Ruth Y. Oliver** [1,2]*, **Carsten Meyer** [3,4,5], **Ajay Ranipeta** [1,2], **Kevin Winner** [1,2], **Walter Jetz** [1,2]*

**1** Department of Ecology and Evolutionary Biology, Yale University, New Haven, Connecticut, United States of America, **2** Center for Biodiversity and Global Change, Yale University, New Haven, Connecticut, United States of America, **3** German Centre for Integrative Biology Research (iDiv) Halle-Jena-Leipzig, Leipzig, Germany, **4** Faculty of Biosciences, Pharmacy and Psychology, University of Leipzig, Leipzig, Germany, **5** Institute of Geosciences and Geography, Martin Luther University Halle-Wittenberg, Halle (Saale), Germany

* ruth.oliver@yale.edu (RYO); walter.jetz@yale.edu (WJ)

**Data Availability Statement:** All supporting data and scripts are available for download at https://github.com/MapofLife/biodiversity-data-gaps associated with the following DOI: https://doi.org/10.48600/MOL-3Y3Z-DW77. National indicator

## Abstract

Conserving and managing biodiversity in the face of ongoing global change requires sufficient evidence to assess status and trends of species distributions. Here, we propose novel indicators of biodiversity data coverage and sampling effectiveness and analyze national trajectories in closing spatiotemporal knowledge gaps for terrestrial vertebrates (1950 to 2019). Despite a rapid rise in data coverage, particularly in the last 2 decades, strong geographic and taxonomic biases persist. For some taxa and regions, a tremendous growth in records failed to directly translate into newfound knowledge due to a sharp decline in sampling effectiveness. However, we found that a nation's coverage was stronger for species for which it holds greater stewardship. As countries under the post-2020 Global Biodiversity Framework renew their commitments to an improved, rigorous biodiversity knowledge base, our findings highlight opportunities for international collaboration to close critical information gaps.

## Introduction

Detection, understanding, and management of global biodiversity change and its manifold consequences [1,2] in a rapidly transforming world rely on comprehensive evidence to establish baselines and assess changes. As discussions of the post-2020 Global Biodiversity Framework of the Convention on Biological Diversity (CBD) enter their final stage, the availability of data and metrics to assess progress toward agreed-upon targets has taken a central role [3–7]. The fundamental need for an improved and shared knowledge base of global biodiversity is recognized in the proposed Target 19, which requires the availability of reliable information on biodiversity status and trends [8].

Descriptions of species' geographical ranges and their temporal dynamics are fundamental biodiversity measures [9], as captured in the species distribution Essential Biodiversity

values are directly accessible for download at mol.
org/indicators/coverage.

**Funding:** This study is supported by the EO Wilson
Biodiversity Foundation, National Science
Foundation grant DEB-1441737 and National
Aeronautics and Space Administration grants
80NSSC17K0282 and 80NSSC18K0435 to W.J. C.
M. acknowledges funding by the Volkswagen
Foundation through a Freigeist Fellowship
(A118199), and additional support by iDiv, funded
by the German Research Foundation (DFG–FZT
118, 202548816). The funders had no role in study
design, data collection and analysis, decision to
publish, or preparation of the manuscript.

**Competing interests:** The authors have declared
that no competing interests exist.

**Abbreviations:** CBD, Convention on Biological
Diversity; GBIF, Global Biodiversity Information
Facility; MOL, Map of Life; SSEI, Species Sampling
Effectiveness Index; SSII, Species Status
Information Index.

Variable [10]. The status and trends of species' geographic distributions are directly related to species' ecological relevance, population size, and extinction risk, and are thus central to the conservation and management of species and their ecological functions [11–13]. Ambitions to limit threats to species and ensure the integrity of ecosystems, which are central goals of the post-2020 Global Biodiversity Framework under discussion [8], critically rely on effective documentation and monitoring of species distributions and changes over time [6,7,14].

Thanks to significant advances in data collection, mobilization, and aggregation [15–17], publicly accessible occurrence data are growing rapidly [9,14,18], with over 1.6 billion occurrence records across sources and taxa available in the Global Biodiversity Information Facility (GBIF). These data represent an increasing array of sources, including museum specimens, field observations, acoustic and visual sensors, and citizen science efforts [19]. Digital platforms such as Map of Life (MOL) have begun to integrate these data through models to bolster a multitude of research and conservation applications [10,20].

Increases in data quantity alone, however, provide little information about overall progress toward an effective spatial biodiversity knowledge base, as records may be highly redundant and cover a limited set of species and regions [21]. Indeed, prior work has revealed significant taxonomic and geographic gaps in the existing data [9,21–26] and highlighted the importance of accounting for expected diversity and scale sensitivity in data coverage assessments [19,21,27–29]. Scientists have identified a range of socioeconomic, linguistic, and ecological drivers for gaps and biases in the current data and identified geographical access, availability of local funding resources, and participation in data-sharing networks as key correlates of data gaps [21,30].

The aforementioned gaps in knowledge highlight the importance of a more informed and coordinated approach to developing an effective spatial biodiversity evidence base. Developing such an evidence base requires metrics that allow changes in biodiversity data coverage over time to inform decision-making. As political units responsible for coordination and stewards of their biodiversity, nations hold the key to incentivizing an improved information base and stand to gain the greatest benefits from broadly improved biodiversity information by enabling monitoring and robust management decisions. For example, the activities of the Mexican National Commission on Biodiversity (CONABIO), a permanent commission of the Mexican federal government, have led to strongly increased biodiversity information in that country that supports conservation decisions in the region [31]. Despite the urgent need to meet international targets and numerous documentations of growing data [32,33], published work has yet to provide quantitative metrics to track nations' progress in closing spatiotemporal biodiversity data gaps [27,34–36].

Here, we provide 2 national indicators in support of the global assessment, monitoring, and decision-support around annual trends in spatiotemporal biodiversity information. These metrics are integrated within a flexible, updatable analytical framework. Specifically, we present and globally implement the MOL Species Status Information Index (SSII), which was developed under the auspices of the GEO Biodiversity Observation Network [37] (https://mol.org/indicators/coverage) in support of IPBES reporting (https://ipbes.net/core-indicators) and global assessment processes [8], as well as the Species Sampling Effectiveness Index (SSEI). We use the indicator framework to compare global and national trends in spatiotemporal biodiversity knowledge since 1950 for over 31,000 terrestrial vertebrate species and over 450 million verified and taxonomically harmonized occurrence records at the level of species, nations, and the globe. We provide a first global assessment for trends in data coverage and sampling effectiveness for terrestrial vertebrates as well as infrastructure to continuously track these indices into the future at MOL (https://mol.org/indicators/coverage).

The SSII quantifies spatiotemporal biodiversity data coverage for a particular grid resolution and species geographic range expectation (Fig 1A). The Global SSII tracks the proportion of expected range cells with records, either for a single species or averaged across multiple species (Fig 1B). The National SSII is calculated using the same method as the Global SSII but is restricted to the range cells inside a particular country (Fig 1B). Steward's SSII follows the National SSII calculation but additionally applies a species-level weight to account for different national stewardships of species (Fig 1B). Nations' varying responsibilities are determined by the portion of a species' global range they hold (e.g., 1 for country endemics; see Fig 1A for illustration and Text A in S1 File for formal description). For a given species, SSII quantifies the proportion of the range with data but not how effectively these data are distributed across the proportion of the range it covers. We characterize sampling effectiveness by relating the realized spatial distribution of records to the ideal uniform distribution based on Shannon's entropy (Fig 1C, Text A in S1 File) normalized to vary between 0 and 1, a metric we call the SSEI. The SSEI is similar to other information theoretic evenness metrics, such as Pielou's index of species evenness, which is also based on normalized entropy [38]. SSEI has the same properties as SSII and can be calculated at the species, national, or global level and additionally can be adjusted by national stewardship for species.

We illustrate the SSII and SSEI for the years 2000 to 2019 for the jaguar (*Panthera onca*) and collared peccary (*Pecari tajacu*), 2 widely distributed species with heterogeneous sampling (Fig 2, Table A in S1 File). The number of records collected annually for the peccary was substantially higher than for the jaguar, ranging from 2- to 10-fold higher data collection (Fig 2A–2C). Subsequently, Global SSII was consistently higher for the peccary than the jaguar, but the difference in values was narrower than the difference in data collection would suggest (Fig 2D).

Such results suggest a much lower sampling effectiveness, as indexed by the SSEI, for the peccary compared to the jaguar, indicating that many peccary records were concentrated in the same regions. SSII improved markedly for the peccary in recent years, reaching 0.03 (i.e., 3% global range cells with annual records). This increase was associated with increasing SSEI, as the number of records collected were only slightly elevated (Fig 2E). National and Steward's SSII calculated for these 2 species was highest in Costa Rica and lowest in Brazil (Fig 2F, Table B in S1 File). National SSEI was generally highest in Brazil and lowest in Colombia (Fig 2G).

## Global and national trends in data coverage and sampling effectiveness

Biodiversity data collection has rapidly proliferated, particularly over the last 2 decades (Fig 3A). However, the proliferation of species records and their translation into biodiversity knowledge has played out along substantially different trajectories among taxa. For example, bird species consistently had the largest number of records, with approximately 1,000-fold greater number of records collected annually and 3-fold greater percentage of expected species recorded compared to other terrestrial vertebrates (Fig 3B). Yet, SSII for birds only exceeded the 3 other groups after 1980 but has since shown near-linear growth in taxon-wide SSII (Fig 3C). Although data collection in terms of number of records for mammal species consistently outpaced that for amphibians and reptiles, data coverage for mammals was lowest in recent years (Fig 3C). Coincident with this rapid rise in data collection and coverage for birds species, however, was a rapid decline in sampling effectiveness (Fig 3D–3F).

Biodiversity knowledge continued to be highly geographically biased over the previous decade (2010 to 2019), with the most complete data coverage found primarily within the United States, Europe, South Africa, and Australia (Fig 4A). Globally, only approximately half

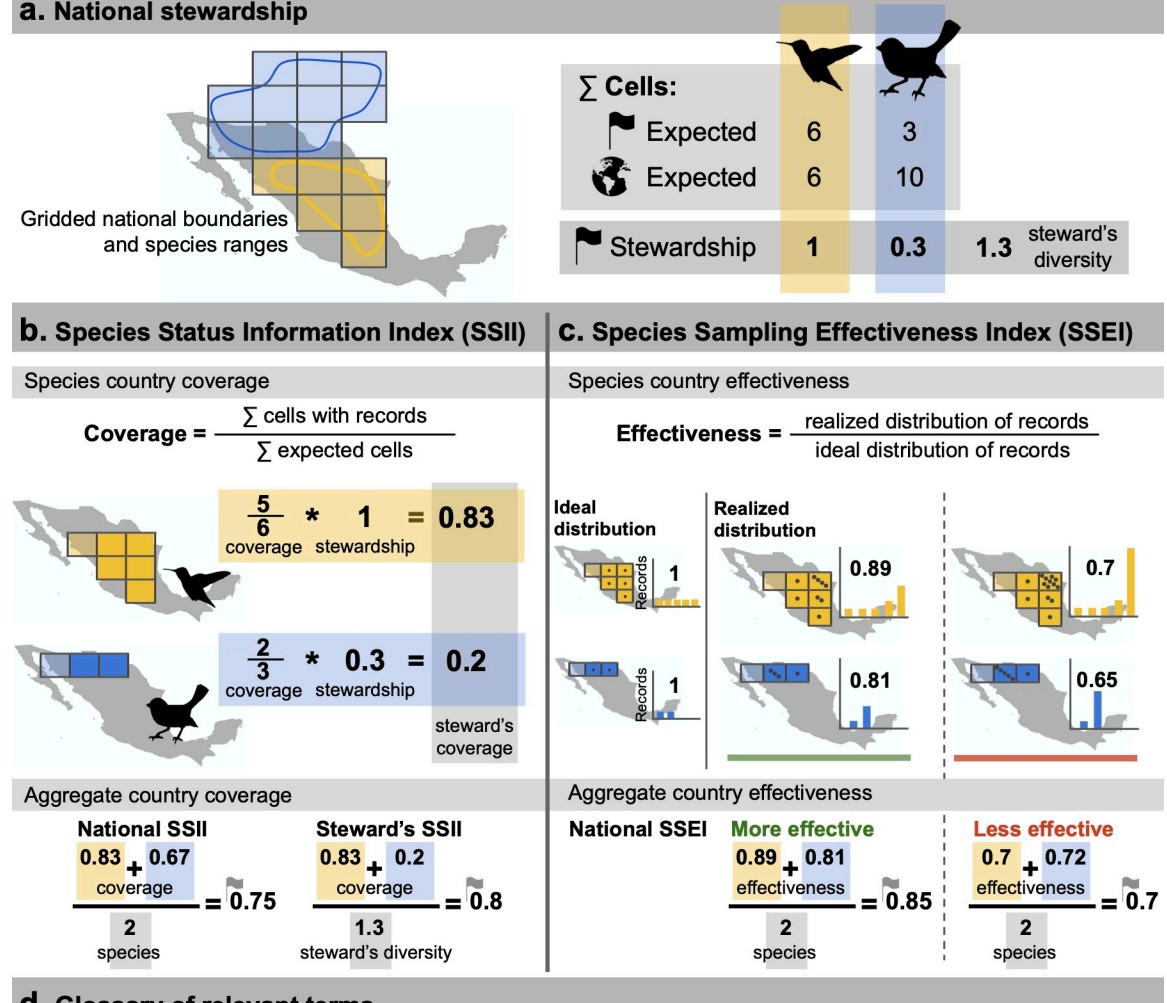

**a. National stewardship**

Gridded national boundaries and species ranges

∑ **Cells:**
🏳 Expected 6 3
🌐 Expected 6 10
🏳 Stewardship **1** **0.3** **1.3** steward's diversity

**b. Species Status Information Index (SSII)**

Species country coverage

$$\text{Coverage} = \frac{\sum \text{cells with records}}{\sum \text{expected cells}}$$

$\frac{5}{6}$ * 1 = **0.83**
coverage  stewardship

$\frac{2}{3}$ * 0.3 = **0.2**
coverage  stewardship

steward's coverage

Aggregate country coverage

**National SSII**
$\frac{0.83 + 0.67}{2} = 0.75$
coverage
species

**Steward's SSII**
$\frac{0.83 + 0.2}{1.3} = 0.8$
coverage
steward's diversity

**c. Species Sampling Effectiveness Index (SSEI)**

Species country effectiveness

$$\text{Effectiveness} = \frac{\text{realized distribution of records}}{\text{ideal distribution of records}}$$

**Ideal distribution** | **Realized distribution**

Records 1 | 0.89 | 0.7
Records 1 | 0.81 | 0.65

Aggregate country effectiveness

**National SSEI** | **More effective** | **Less effective**

$\frac{0.89 + 0.81}{2} = 0.85$
effectiveness
species

$\frac{0.7 + 0.72}{2} = 0.7$
effectiveness
species

**d. Glossary of relevant terms**

**Species Status Information Index (SSII)** - captures how well available data addresses species' expected distributions.

**Species Sampling Effectiveness Index (SSEI)** - quantifies how evenly available data are distributed.

**National SSII and SSEI** for a given species group takes the mean across species expected in a country.

**Steward's SSII and SSEI** adjusts the mean across species based on national stewardship of species (the proportion of global expected distribution it holds.

**Map of Life (MOL)** - integrates spatial biodiversity and environmental data through models to provide best-possible information on global species occurrences and their change (http://mol.org).

**Global Biodiversity Information Facility (GBIF)** - an international network and data infrastructure that serves as an aggregator of species point occurrence data (http://gbif.org).

**Fig 1. SSII and SSEI metrics of biodiversity data coverage and effectiveness.** The metrics are illustrated for 2 hypothetical species with geographic range delineated by binary (e.g., expert range) maps and are assessed for an example 110-km equal-area grid. (**a**) National stewardship of species is calculated based on the relative portion of species' ranges falling inside a country. (**b**) At the species level, the SSII is given as the proportion of cells expected occupied with records in a given year. In this hypothetical example, coverage is 0.83 and 0.67 for species where 5 out of 6 and 2 out 3 expected grid cells have data. Steward's SSII adjusts this coverage by their respective national stewardship (0.83 and 0.2). Species-level SSII can be aggregated to the national level via 2 formulations. National SSII for a given taxonomic group takes the mean coverage across all species expected in a country (0.75). Steward's SSII adjusts the mean coverage across species by their respective national stewardship (0.8). (**c**) SSEI compares the entropy of the realized distribution of records to that of the ideal distribution (see Text A in S1 File), where uneven sampling (lower SSEI) is considered less effective than more even sampling (higher SSEI). National SSEI takes the mean across all species expected in a country. (**d**) Glossary of relevant terms. *Artwork from plylopics.org (see Text A in S1 File)*. GBIF, Global Biodiversity Information Facility; MOL, Map of Life; SSEI, Species Sampling Effectiveness Index; SSII, Species Status Information Index.

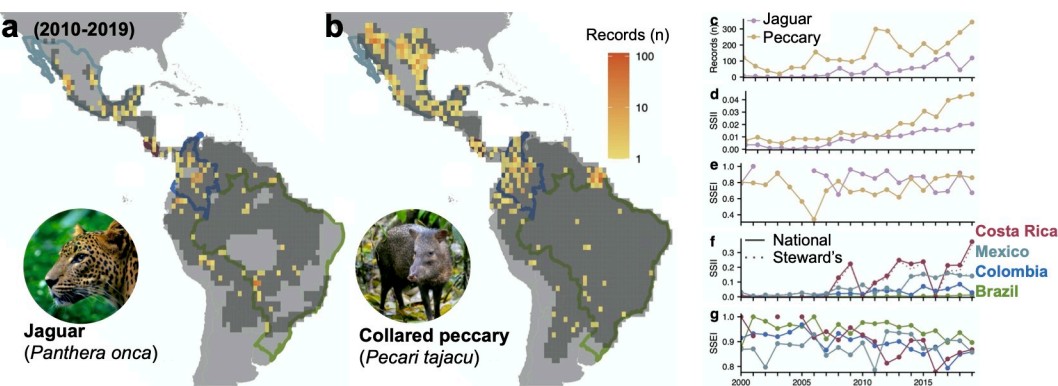

**Fig 2. Species and national example patterns and trends.** SSII and SSEI trends illustrated for 2 species, the jaguar (*Panthera onca*) and collared peccary (*Pecari tajacu*). (**a, b**) The expected occupied cells are shown in dark gray, and total number of records collected 2010–2019 in color. (**c–e**) Species-level time series of the total number of records (**c**), Global SSII for the whole species range (i.e., all countries with expected range) (**d**), and Global SSEI (**e**) across their expected range. (**f, g**) Resulting National and Steward's SSII (**f**) and SSEI (**g**) for 4 countries. *Photographs from Wikimedia (see Text A in S1 File). National boundaries from gadm.org. Numerical values available in Tables A and B in S1 File. The data underlying this figure may be found in https://mol.org/indicators/coverage and https://github.com/MapofLife/biodiversity-data-gaps.* SSEI, Species Sampling Effectiveness Index; SSII, Species Status Information Index.

of nations (42%) showed increasing, significant trends ($p < 0.01$) in coverage averaged across taxa over the previous decade (Fig 4B). For those nations showing increases, trends are driven primarily by the rapid increase in avian distribution data (Fig 4C, Fig A and Table C in S1 File). Nearly half of nations (47%) showed significantly increasing data coverage for birds, whereas less than 20% of nations had increasing trends for other taxa (Fig 4C). This suggests that despite increasing data availability for all taxa, a majority of nations are not making progress in closing information gaps for mammals, amphibians, and reptiles.

Interestingly, several world regions that historically most comprehensively sampled the full suite of local species across their geographic ranges are no longer continuing along increasing trajectories (Fig 4B). For example, Western Europe, South Africa, and Australia appear to have slowed in their coverage progress (i.e., SSII across taxa), possibly reflecting challenges in the continued mobilization of existing datasets or a lack of impetus to engage in new initiatives [39]. However, we anticipate that even under constant effort, nations' coverage may asymptote as marginal gains become more challenging to achieve. Therefore, asymptoting trajectories in data coverage may suggest that nations are operating at maximum capacity. Thus, nations with slowing trends may best contribute to CBD goals by partially shifting their investments in national biodiversity data creation toward supporting targeted data mobilization and capacity-building in nations that have so far lagged behind through direct partnerships [40]. By contrast, much of Asia, South America, and Western and Northern Africa had increasing coverage over the previous decade from initially low values, suggesting encouraging information prospects if trends continue (Fig 4B). Our results underscore the importance of regionally targeted capacity-building and data mobilization initiatives that support regions with historically limited data coverage. Such efforts currently underway include GBIF's Biodiversity Information Fund for Asia and Biodiversity Information for Development program focused in sub-Saharan Africa, the Caribbean, and the Pacific.

Despite the astounding accumulation of biodiversity records, not all data have translated to new knowledge of species distributions [41]. While potentially useful for other ecological applications, sampling effectiveness of biodiversity data as indexed by the SSEI varied considerably among nations over the previous decade, with lowest effectiveness typically within

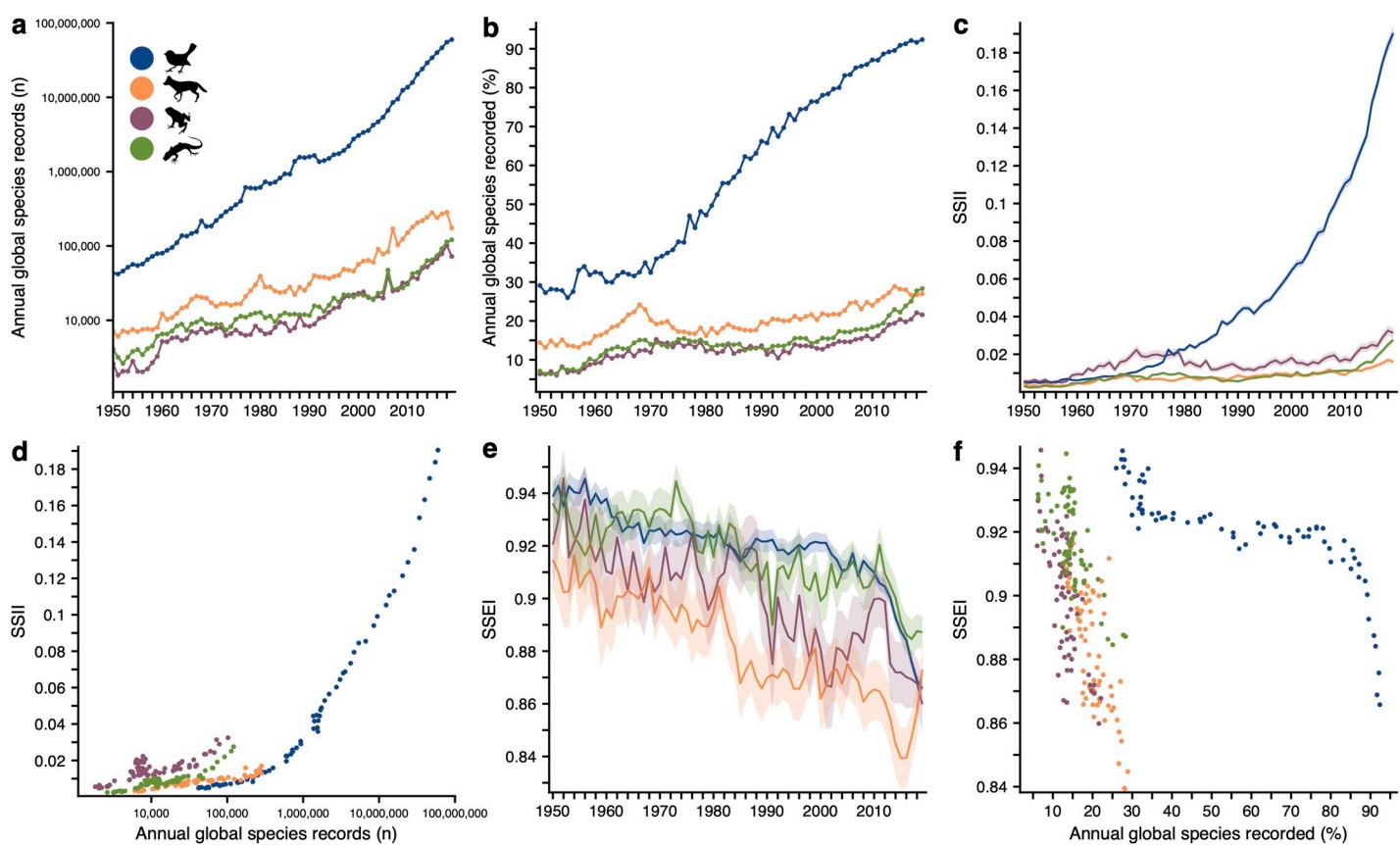

**Fig 3. Global trends data coverage and sampling effectiveness across 4 terrestrial vertebrate groups. (a–c)** Trends in total annual record counts (**a**), percentage of expected species recorded (**b**), and the Global SSII (**c**). Global SSII is based on data coverage across species' ranges without consideration of national boundaries. Alternatively, Global SSII for a species is the sum of Steward's SSII across the nations where it is expected to occur. (**d**) Relationship between annual total record counts and Global SSII. (**e**) Trends in Global SSEI. (**f**) Relationship between percentage of expected species recorded and Global SSEI. (**c, e**) Lines and shading represent means and 95% confidence intervals across species within classes. (**d, f**) Relationships are shown over the past 70 years (1950–2019). Colors in **a–f** indicate birds (blue), mammals (orange), amphibians (purple), and reptiles (green). *Artwork from phylopics.org (see Text A in S1 File). The data underlying this figure may be found in https://mol.org/indicators/coverage and https://github.com/MapofLife/biodiversity-data-gaps.* SSEI, Species Sampling Effectiveness Index; SSII, Species Status Information Index.

Western Europe, North America, and Australia (Fig 4D). National trends in effectiveness also appear to be largely driven by the trends in sampling effectiveness for bird species, which has declined rapidly over the past 2 decades, constraining the direct conversion of the immense accumulation of data into data coverage (Fig 4E and 4F).

Differences and trade-offs in data coverage and effectiveness among taxa appear to be largely driven by the way in which data are collected for different taxonomic groups. As of 2016, nearly all records for birds in GBIF (>90%) came from direct observations, as opposed to museum specimens that constituted the primary source of records for amphibians, reptiles, and, to a lesser degree, mammals [26]. However, these differences in sources are likely to narrow as citizen scientist programs not restricted to birds continue to grow in popularity (e.g., iNaturalist). SSII for birds did not surpass that for other classes until the 1980s, despite having an order of magnitude greater number of records. Further, for the same number of records, birds had the lowest SSII among terrestrial vertebrates and appear to only have achieved the highest SSII through sheer volume of records, as opposed to strategic sampling (Fig 3D). Although data coverage for birds increased throughout much of the 20th century, the launch of citizen science platforms such as eBird [42] in the early 2000s undoubtedly played a large

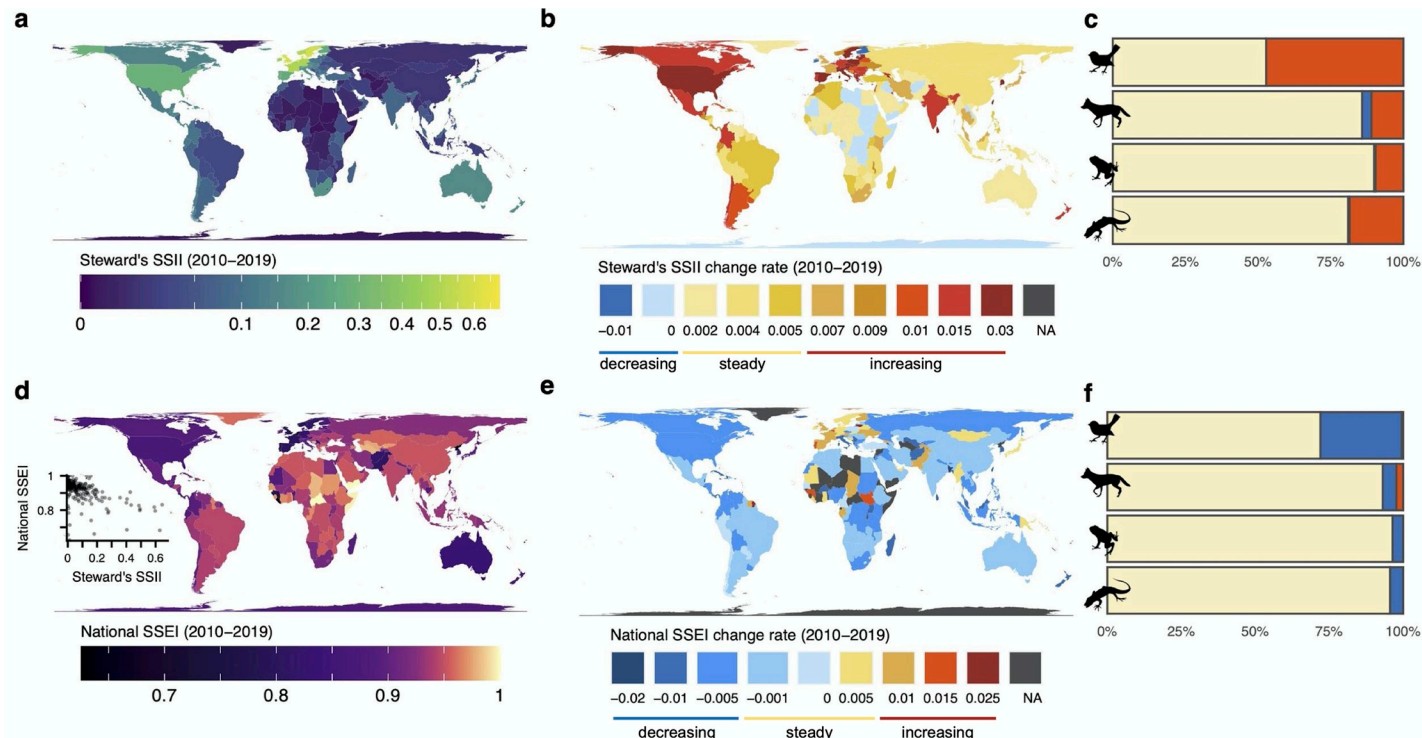

**Fig 4. National patterns and trends in spatial biodiversity data coverage and sampling effectiveness.** (**a, d**) Mean Steward's SSII (**a**) and National SSEI (**d**) over the previous decade (2010–2019) averaged across terrestrial vertebrates; the relationship between data coverage and sampling effectiveness is shown as inset. (**b, e**) Change rate in Steward's SSII (**b**) and National SSEI (**e**) over the previous decade. Maximum values for each color bin are labeled below each map. (**c, f**) Percentage of nations with no significant ($p < 0.01$) trends (beige) and significant decreasing (blue) or increasing (red) trends in Steward's SSII (**c**) and SSEI (**f**) over the previous decade for birds, mammals, amphibians, and reptiles. *Artwork from phylopics.org (see Text A in S1 File). National boundaries from gadm.org. The data underlying this figure may be found in https://mol.org/indicators/coverage and https://github.com/MapofLife/biodiversity-data-gaps*. SSEI, Species Sampling Effectiveness Index; SSII, Species Status Information Index.

role in the expeditious increase in coverage [19]. However, this onslaught of observations has not been maximally leveraged to enhance the global biodiversity information base, as seen in the coincident decline in avian sampling effectiveness (Fig 3E).

The accelerated pace of data coverage for birds compared to other vertebrate groups points to the tremendous role that non-museum–based data collection can play in closing knowledge gaps [43–45]. However, the rapid decline in sampling effectiveness we found for bird species, coincident with the growth in citizen science platforms, suggests that these data have not collected to optimally support closing knowledge gaps. While the contributions of citizen science have been invaluable, expanding the impact of citizen science initiatives for information growth will likely benefit from initiatives and guidance addressing the most effective and complementary contributions (i.e., addresses undersampled species or regions) [46,47]. The rapidly changing landscape of citizen science initiatives will need to be complemented by further supporting and growing coordinated programs through international organizations or government agencies that ensure improved data coverage. Citizen science platforms could shift incentives from numbers of records collected or species identified to the value of records contributed. Quantifying and identifying particularly important data contributions through products such as the SSII and SSEI, which can be updated and delivered through the MOL infrastructure, can support naturalists and initiatives to fill key geographic and taxonomic gaps.

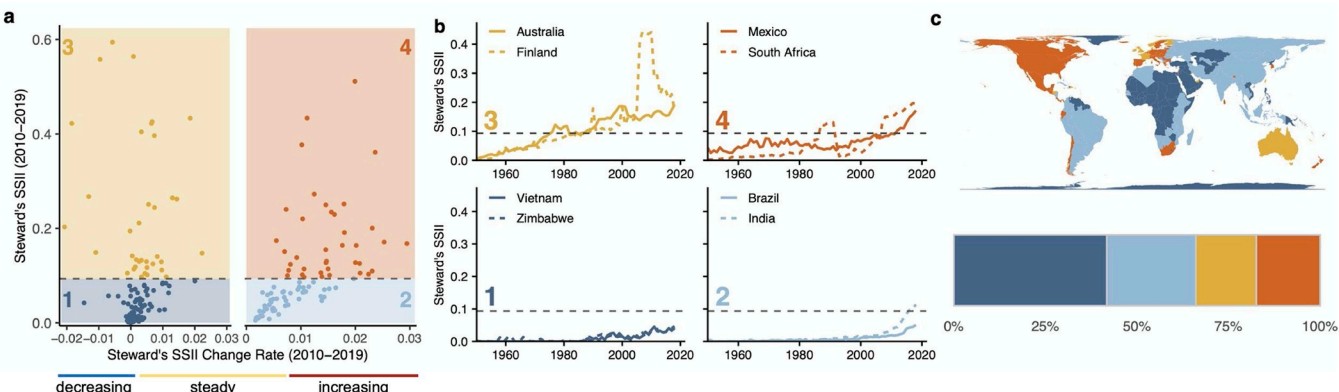

**Fig 5. Typologies of nations' data coverage and trends. (a)** Mean values and change rates in Steward's SSII over the previous decade (2010–2019). Horizontal dashed line represents the global mean of Steward's SSII. Left panels show nations with no significant or decreasing trends in coverage. Right panels show nations with significant ($p < 0.01$) increasing trends in coverage. We categorized nations into the following 4 main types based on Steward's SSII status and trends over the previous decade: (1) coverage **less than** the global mean with **no or decreasing** trend (2010–2019) (42% of nations); (2) coverage **less than** the global mean with an **increasing** trend (24%); (3) coverage **greater than** the global mean with **no or decreasing** trend (17%); and (4) coverage **greater than** the global mean with an **increasing** trend (17%). **(b)** Example time series for nations within each type. **(c)** National assignment to quadrants. Bar plot shows percentages of nations within each quadrant. *National boundaries from gadm.org. The data underlying this figure may be found in https://mol.org/indicators/coverage and https://github.com/MapofLife/biodiversity-data-gaps.* SSII, Species Status Information Index.

## Typologies of national monitoring efforts

National biodiversity monitoring is influenced by a myriad of social, political, economic, and geographic factors [21,22,30,48,49]. We categorized nations into the following 4 main types based on Steward's SSII status and trends over the previous decade: (1) coverage **less than** the global mean with **no or decreasing** trend (2010 to 2019) (42% of nations); (2) coverage **less than** the global mean with an **increasing** trend (24%); (3) coverage **greater than** the global mean with **no or decreasing** trend (17%); and (4) coverage **greater than** the global mean with an **increasing** trend (17%) (Fig 5A). We highlight national trajectory examples from each group (Fig 5B). Status and trends in Steward's SSII differed strongly among continents (Fig 5C).

Biodiversity data coverage within Mexico has followed a strong, and increasing, trajectory in both the 20th and 21st centuries. Despite lower coverage through periods of the 20th century, South Africa has had similarly strong and increasing data coverage over the previous decade. Many nations that had historically limited data coverage showed recent increases in coverage, for example, Brazil. These trajectories in data coverage may be due to political decisions and national infrastructure, which supports biodiversity data collection and mobilization. For example, the establishment of a national biodiversity program (CONABIO) [31] and large-scale atlasing efforts, such as the Southern African Bird Atlas Project [50,51].

Through their national commitment to the CBD targets to decrease species extinctions, nations are asked to monitor the species for which they hold greatest responsibility, or, in the case of endemic species, full responsibility. By comparing National and Steward's SSII, we found that a majority of nations (50%) preferentially survey species for which they hold a high proportion of the global ranges (Fig B in S1 File). This may reflect a tendency for endemic biodiversity to confer special cultural importance and for societal interests to influence research agendas [49] or simply reflect the preferences of citizen scientists aiming to boost their life lists. Selective monitoring based on nations' stewardship of species may beneficially promote conservation agendas within nations that have primary control of habitats that species rely on. With this goal in mind, our analysis highlights when nations fall behind on sampling species

for which they have high stewardship and thus play a particularly large role in species' conservation (e.g., Indonesia and Costa Rica).

## Future directions for tracking global biodiversity knowledge

We recognize the limitations of SSII, or any single metric of biodiversity data coverage, to address the range of research and monitoring needs. Our formulation assumes a specific set of taxonomic, spatial, and temporal units and places a burden on nations with particularly high diversity or large national areas to achieve high scores. Furthermore, the annual time units and a relatively coarse grid for the SSII patterns presented here are insensitive to spatiotemporally dense data that could reveal seasonal dynamics and additional insights offered by repeat samples (e.g., in occupancy modeling frameworks) [52]. Similarly, by penalizing uneven sampling, the SSEI ignores applications that require repeat sampling. In its current form, the SSII also does not account for coverage in environmental space (e.g., as relevant for model-based inference and Essential Biodiversity Variable production) [10]. Further, because the SSII is currently based on static representations of species ranges, it does not capture range dynamics, such as in new invasions or range shifts [9]. This could be addressed through timely updates of range expectations or other invasion-specific information [53]. Dynamically tracking species distributions will be particularly important in cases where species ranges shift across national boundaries, resulting in new monitoring responsibilities. Our methodology and analysis infrastructure is capable of flexibly accommodating different spatial resolutions (Fig C and Text A in S1 File) as more precise information on species' ranges becomes available (e.g., through species distribution modeling) for a broader range of taxa.

A group of alternative approaches to quantify sampling completeness rely on parametric or nonparametric richness estimates based on extrapolation of assemblage species accumulation curves [27,34–36]. These approaches provide an important complementary contribution especially for extremely undersampled or underdescribed taxa where globally comprehensive species range expectations, which are necessary for SSII calculation, remain unavailable. However, richness estimates from extrapolation approaches can vary dramatically with the specific methodology used and structure of input data. As such, there are competing recommendations for their development and use [54–56]. The SSII avoids potential pitfalls and the limited transparency of extrapolation approaches by relating record collection directly to best-possible species-level expectations. Therefore, the SSII allows for decision support at the species level, which is not possible with extrapolation approaches. While this study is limited in scope to terrestrial vertebrates, the framework is easily extended to address other taxonomic groups and realms with ongoing, comprehensive distribution mapping efforts, such as plants and certain marine and invertebrate groups [18,57]. The SSII offers an effective initial characterization of biodiversity information at the species, national, and global scales, with the potential to extend the metric to account for different spatiotemporal grains (Fig C in S1 File), taxa, and data types.

## Conclusions

The framework and indicators presented here offer a quantitative and comparable characterization of species, national, and global trajectories in closing biodiversity information gaps. The need for more comprehensive quantitative and standardized biodiversity information to support policy and action not only underpins improved Essential Biodiversity Variables [10] but is also recognized as critically needed in recent assessments of IPBES and the post-2020 Global Biodiversity Framework. Our findings suggest that trends in data coverage fundamentally differ by taxa and region and highlight the need to complement and reassess biodiversity

sampling strategies to most effectively translate data collection into biodiversity knowledge useful for management and decision-making.

## Supporting information

**S1 File. Text A.** Methods, Supplementary Text, and Supplementary Acknowledgments to support main text. **Fig A. National patterns in data collection, coverage, and sampling effectiveness (2010–2019). (a)** Change rates in Steward's SSII and National SSEI. Dashed lines represent zero slopes. **(b, c)** Relationship and mismatch between Steward's SSII and total spatiotemporal records collected nationally **(b)** and the percentage of expected species nationally recorded **(c)**. **(d)** Relationship between the percentage of expected species nationally recorded and mean National SSEI. *The data underlying this figure may be found in https://mol.org/indicators/coverage and https://github.com/MapofLife/biodiversity-data-gaps*. **Fig B. National stewardship in data coverage. (a)** National and Steward's SSII over the previous decade (2010–2019). Points are colored by the percent difference between National and Steward's SSII. Dashed line represents the 1:1 line between variables. **(b)** Relative stewardship of nations, as estimated by percent difference, over the previous decade. Color scale matches that in panel **(a)**. *National boundaries from gadm.org. The data underlying this figure may be found in https://mol.org/indicators/coverage and https://github.com/MapofLife/biodiversity-data-gaps* .

**Fig C. Empirical demonstration of the effects of spatial resolution on the SSII and SSEI. (a, b)** Thresholded species distribution model output (Ellis-Soto and colleagues, 2021) rescaled to 3 spatial resolutions (110, 55, and 27.5 km) for 2 hummingbird species, **(a)** the Glowing puffleg (*Eriocnemis vestita*) and **(b)** White-sided hillstar (*Oreotrochilus leucopleurus*). Grid cells are colored by the number of records collected between 2000–2019. **(c)** Annual SSII (solid lines) and SSEI (dashed lines) computed at 3 spatial resolutions. **(d–i)** Comparison of SSII (d–f) and SSEI (g–i) values among spatial resolutions (**d, g**: 100 vs. 55 km; **e, h**: 55 vs. 27.5 km; **f, i**: 110 vs. 27.5 km). Gray shading shows 95% confidence interval. Colored text displays slope estimates and 95% confidence intervals for each species (blue: *Eriocnemis vestita*; green: *Oreotrochilus leucopleurus*). *The data underlying this figure may be found in https://mol.org/indicators/coverage and https://github.com/MapofLife/biodiversity-data-gaps*. **Fig D. Theoretical examples of the Species Sampling Effectiveness Index (SSEI).** Each line corresponds to theoretical cases with different levels of evenness of the distribution of biodiversity records for an idealized species with the same range size. In these examples, the proportion of the sampled range with a single record vs. alternate values (1, 2, 10, 100, and 1,000) is adjusted from 0 to 1. SSEI is highest in cases with uniform or near-uniform sampling (i.e., all grid cells either contain 1 or 2 records). SSEI is lowest in cases with highly uneven sampling (i.e., a mixture of grid cells with either a single record or 100–1,000 records). These examples also highlight that SSEI is identical in the cases where redundant sampling is uniform (i.e., values are the same if all cells have a 1, 10, or 1,000 records). Additionally, SSEI approaches the maximum value when only a small minority of cells contain more than a single record (i.e., the proportion of cells with a single record >90%). **Table A. Species example coverage and sampling effectiveness values.** Values presented for the jaguar (*Panthera onca*) and collared peccary (*Pecari tajacu*) as demonstrated in Fig 2C–2E. *The data underlying this table may be found in https://mol.org/indicators/coverage and https://github.com/MapofLife/biodiversity-data-gaps*. **Table B. National example data coverage and sampling effectiveness values.** Values presented for the jaguar (*Panthera onca*) and collared peccary (*Pecari tajacu)* as demonstrated in Fig 2F and 2G. *The data underlying this table may be found in https://mol.org/indicators/coverage and https://github.com/MapofLife/biodiversity-data-gaps*. **Table C. National data coverage and sampling effectiveness values over the previous decade (2010–2019).** ISO3 codes and mean values for National

and Steward's SSII and SSEI for nations. *The data underlying this table may be found in https://mol.org/indicators/coverage and https://github.com/MapofLife/biodiversity-data-gaps.* (PDF)

## Acknowledgments

We thank the Map of Life team for their support and expertise, particularly Vijay Barve, Yanina Sica, and Michelle Duong. We are also grateful to the GBIF team, specifically Joe Miller, Tim Hirsch, and Tim Robertson for their help with data and manuscript feedback.

## Author Contributions

**Conceptualization:** Ruth Y. Oliver, Carsten Meyer, Walter Jetz.

**Data curation:** Ajay Ranipeta.

**Formal analysis:** Ruth Y. Oliver.

**Investigation:** Ruth Y. Oliver, Walter Jetz.

**Methodology:** Ruth Y. Oliver, Carsten Meyer, Ajay Ranipeta, Kevin Winner, Walter Jetz.

**Project administration:** Walter Jetz.

**Resources:** Walter Jetz.

**Supervision:** Walter Jetz.

**Visualization:** Ruth Y. Oliver.

**Writing – original draft:** Ruth Y. Oliver.

**Writing – review & editing:** Ruth Y. Oliver, Carsten Meyer, Ajay Ranipeta, Kevin Winner, Walter Jetz.

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
