## [Editor Report · Decision Letter 0]

13 Nov 2020

Dear Dr Oliver, 

Thank you for submitting your manuscript entitled "Global and national trends in documenting and monitoring species distributions" for consideration as a Research Article by PLOS Biology.

Your manuscript has now been evaluated by the PLOS Biology editorial staff, as well as by an academic editor with relevant expertise, and I'm writing to let you know that we would like to send your submission out for external peer review.

Please re-submit your manuscript within two working days, i.e. by Nov 17 2020 11:59PM.

Kind regards,

Roli Roberts

Senior Editor

PLOS Biology

---

## [Decision Letter · Decision Letter 1]

19 Jan 2021

Dear Dr Oliver,

Thank you very much for submitting your manuscript "Global and national trends in documenting and monitoring species distributions" for consideration as a Research Article at PLOS Biology. Your manuscript has been evaluated by the PLOS Biology editors, an Academic Editor with relevant expertise, and by four independent reviewers. Many thanks for your patience over the holiday period.

You’ll see that the assessments are broadly very positive, but there are multiple requests for you to justify and/or conduct sensitivity analysis on the choice of grid size. There are several further substantial requests, regarding treatment of empty cells, treatment of migrant species, data and code availability (please see PLOS' Data Policy, which is quite stringent), methodological clarifications and statements of limitations. Reviewer #4 also feels that the SSEI metric may be flawed and not truly independent from the SSII, making it redundant over quite a swathe of parameter space; you hould address or rebut this. The Academic Editor asked me to draw your attention to points 1 and 2 from reviewer #1, and the multiple calls for data availability (which we will enforce).

In light of the reviews (below), we will not be able to accept the current version of the manuscript, but we would welcome re-submission of a much-revised version that takes into account the reviewers' comments. We cannot make any decision about publication until we have seen the revised manuscript and your response to the reviewers' comments. Your revised manuscript is also likely to be sent for further evaluation by the reviewers.

We expect to receive your revised manuscript within 3 months. 

**IMPORTANT - SUBMITTING YOUR REVISION**

*Re-submission Checklist*

*Published Peer Review*

*PLOS Data Policy*

*Blot and Gel Data Policy*

Sincerely,

Roli Roberts

Senior Editor,

rroberts@plos.org,

PLOS Biology

REVIEWERS' COMMENTS:

Reviewer #1:

It was a pleasure to read the manuscript titled "Global and national trends in documenting and monitoring species distributions". I actually saw this posted as a preprint, so getting the opportunity to review it afforded me to take the time and dive in more than I would have as it was in my 'to read' pile.

Overall, I fully support the notion behind what the authors are highlighting here. This is best captured by the idea that 'more data' does not necessarily mean more informed biodiversity decisions and policies (line 61). We are getting more data now than ever before, but are we any better off for it? Who knows! I agree that the general approach to provide quantitative metrics to track a given nation's progress is important. I also think that this could be a method for countries to hold one another accountable in terms of their contributions to biodiversity sampling. I think this paper is suitable for PLoS Biology, and will generate broad interest for an international audience no doubt. However, I think there are a few concerns that if addressed, could significantly strengthen this manuscript. Here I have a few 'major concerns' and then some minor concerns after this. I admit these are somewhat substantial concerns/comments/requests. But in all honesty, I think this has major potential, so would hate to see it published without highlighting the full potential of what you are doing. It will be interesting to see what other Reviewers think about this one, but regardless, I hope these comments are helpful for you!

1) Although I fully support the notion here and behind this paper, I am on the fence about this paper as it currently stands. This is because it currently reads as 50% a methods paper and 50% interpretation of the results. I think the SSII and SSEI are interesting and novel, but the derivation is buried in the methods and not given proper thought at times (e.g., in regard to potential biases that are mentioned in passing). And because of this, I think the manuscript suffers because I am left wanting more of the method development and showing that it truly will work, but also left wanting more of the interpretation and what countries are doing better or worst etc. etc. This is highlighted most in line 240. I really think this would be much stronger if the authors could show that indeed, the metric can be extended at different spatial grains. I would recommend a case study of a few species and going down by the size of the grid, as my current worries is that the 110 km is just far too large for a lot of species. (See next point).

2) I kind of see why the ideal distribution is equal sampling among grid cells. But then this opens up a lot of questions about EOO and AOO, and how the SSII performs in relation to these. The assumption that grid cells are empty then the species doesn't exist there, is not great, and potentially a fundamental flaw as the manuscript currently stands. But I understand you don't get 'absence' information. I think future work would be able to improve these indices by inferring absences based on number of records etc. - especially at large spatial scales. The authors mention this limitation towards the end of the manuscript. But, I think if the authors can demonstrate that this isn't a big issue. Maybe with the jaguar and peccary. Then this would really strengthen the manuscript. It would also be informative to know when this method works for some species as opposed to others

3) What about migrants? This is mostly related to birds. But as I understand, you take all records regardless of time of year and integrate them spatially with grids. But, this could influence the country's reporting as their 'range' only applies to a certain time of year. E.g., neotropical migrants. In the methods the authors argue that by taking the average over many species, some idiosyncrasies among species likely are minimal. But it would be really worth showing this empirically, instead of just making the conceptual argument. Perhaps you could use migrants as a case study.

4) Data availability: According to the version of the manuscript I received, the data availability statement is as follows: "All data is available at the Map of Life (www.mol.org).". I'm a big fan of MOL, but I don't know how to get data our of there. I don't see any type of 'download data' button or tabs. Do I need an account? I just spent about 20 minutes trying to download some datasets, but didn't get anywhere. Making an account is prohibitive, and given the push for open access data, this shouldn't be a necessity. I even tried tracking down the GitHub repo (https://github.com/MapofLife/indicators), but didn't see any actual data here. Given the requirement to publish in PLoS of all data underlying the findings being "fully available", I would hope that the authors can make these data available properly. Even if you can actually get them from MOL and I'm just dense in my ability to do so, I still think as a stand alone paper these data should be available so that the results and findings are reproducible. That is, the data from 1950-2019 for each species, and the SSII and SSEI for each species, as well as probably separate datasets for the country-level averages of these, should be made available in a repository that is permanently archived before acceptance of this manuscript.

Minor comments:

If PLoS allows it, I would suggest using some subheadings in the main text. There are a few jumps to new paragraphs where the flow is not great, and I think subheadings could help alleviate this.

Lines 60-61: Totally agree!

Line 69-74: I don't doubt these statements at all. I'm just wondering if there is any references/qualitative research surrounding this? Mainly, because if there is, it would be cool to highlight it. I suspect such references would be hard to come by, however. But perhaps at the very least, the authors could provide an example of a few nations where they incentivized an improved information base and this was led from a national planning coordination? This is discussed a little bit when the results are interpreted, but an example here would help sell this important point a bit more.

Line 81: "advance and globally implement". So is this method used previously? Has it been published before? Or is what you are doing now, the first time it is being described? This is related to my main point above. If it was published through geo bon, to what extent?

Lines 99-100: Here you say, "not just by the number of records, but also by how effectively the records cover a species' full geographic range". But, then what is the actual difference between SSII and SSEI in words? Or is SSEI just a component of the SSII? Perhaps removing this word 'effectively' and rephrasing will help show that there is indeed a difference between SSII and SSEI?

Line 138: This, to me, is the most likely scenario for Western Europe, South Africa, and Australia. Surely they haven't slowed in their coverage progress, necessarily? But just they are operating at maximum ability?

Lines 140-142: This is interesting! And I agree. But incredibly political. If I understand correctly, the authors are highlighting that, for example, Australia should begin to work on capacity-building in the Pacific Islands and/or Southeast Asia? This is way easier said than done, but I get the point by the authors.

Line 162: It is funny to think of this reference as 'outdated', but they downloaded data from GBIF in 2016, and only used 649 million occurrences (< half of what is in GBIF now). I would bet that the proportion of data from museum specimens has drastically decreased by now in 2020, mainly due to programs like iSPOT and iNaturalist growing in popularity and being integrated into GBIF.

Lines 178-181: I think this is great and I agree!

Lines 203-207: Interesting!!

Line 565: Are these range maps available somewhere in a repository? Can they all be downloaded from MOL for instance? This is core to your analysis, so it would be good to highlight this, although I wouldn't suggest you have to make these available.

Line 578: What about duplicates in GBIF? There are a lot, often in GBIF. https://recology.info/2016/03/scrubr/

Line 617-624: See general comments above.

Line 626-642: ++1 for using geohashes. The field of ecology rarely uses these! Some information here about where your analyses were performed would be handy. Was it using BigQuery (I'm guessing)? Or qGIS?

Lines 712-715: I am not fully following this part. Could you elaborate please.

Lines 830-849: Presumably these will be made available in csv format or some other format for readers? Also, see comments on data availability above.

Reviewer #2:

I enjoyed the introduction it contained a lot of good relevant background information that will be easy to digest for non-experts.

Given the dearth of data at a global scale it is very difficult to identify indicators that will be meaningful a large scale. The SSII and SSEI show promise. SSII quantifies spatiotemporal data while SSEI adds an adjustment for sampling effectiveness. A strength of SSII is indicated by the best sampled group where birds SSII has increased but sampling effectiveness has decreased due to the redundant sampling of citizen science efforts.

Most of the discussion is about the avian results which makes sense due to the higher level of data available for birds. It would be useful to have more text about other taxa. From figure 4 there is less applicability for non-bird data. Is it useful at all for those taxa, specifically reptiles and amphibians?

The limitations section is appreciated. To strengthen your argument for use of these metrics, I would like to hear clear recommendations from the authors to make the metrics more valuable. There clearly is not sufficient data for global recommendations to track some specific taxa to give global coverage, maybe someday. It appears that focused, periodic monitoring of endemic taxa would be the best use of resources, especially at the national level. Since data lack for most groups, can the authors make any recommendations of a subset of taxa that would be useful for nations to track?

Fig 1b: It is unclear to me where the numerators of the aggregate country cover calculations come from.

Starting at line 121: This starts out as a discussion of birds but it is unclear from the second half of the paragraph if it is still just birds or all data in the figures. This needs clarification and separate paragraphs to help the reader.

Line 130: clarify does the 42% refer only to birds. Please add birds to clarify.

Lines 134-136: which figure does this refer too? Any comments on the non-avian portion of the figure?

Figure 5 legend error? Quadrant definitions 1 and 3 as well as 2 and 4 are identical. It is correct in the text, line 185. Please consider changing the color of Fig 5a: light blue for background for quadrant is confusing. It printed out nearly identical blues in my printout.

Line 192: I am not convinced by the discussions correlating trajectories with political decisions and national infrastructures changes.

Paragraph 201 is strong and will be useful for the community to hear.

Would a sliding window view of the data be feasible? The decade level analysis is necessary due to data but would the sliding window show trends better?

Reviewer #3:

[identifies himself as Dominique G Roche]

I was pleased to review PBIOLOGY-D-20-03303_R1 "Global and national trends in documenting and monitoring species distributions". Overall, I think that the research has considerable value. Reducing biodiversity information gaps is a global priority and the proposed framework appears sound. That being said, I am no expert in biodiversity conservation and the relevance/novelty of the indices proposed should be assessed by reviewers with greater expertise in this field. My main criticism of the manuscript is that the text and some of the concepts were difficult to understand at times. Many sentences would benefit from being more clearly written and the figure/table captions were often confusing and/or lacked sufficient information. The figures are critical for readers to understand the content of this manuscript - greater effort should be invested in ensuring that the captions clearly walk the readers through the results.

It would be helpful if the authors included a glossary as a text box to prominently define/explain key terms such as GBIF, MOL, Global/National/Stewards' SSI, SSEI. I think this addition would greatly enhance the ease with which the paper could be read and its accessibility to a broad readership.

The authors mention the importance of scale in the introduction (L64) but there is no justification for the grid size of 110 x 110 km used in the study other than it was "the finest spatial grain appropriate" (L569). There is also no discussion of the potential consequences of using a different scale in their analysis.

The explanation of how 'expected occupied cells' were determined is limited (L587-590).

Few of the test statistics mentioned in the methods (L683-688) are reported in the manuscript. P-values alone are often presented (L722-769) or relationships and differences are only referred to as 'significant' or 'non-significant'.

The manuscript lacks a data availability statement. It is not sufficient to state (in the metadata) that the data are "available at the Map of Life (http://www.mol.org)". The exact link to the specific data used in the study should be provided for reproducibility purposes. Ideally, the data would be downloaded from mol.org and archived in a trusted repository, unless the authors can demonstrate that MOL is funded for the next 50 years (as is the case for trusted repositories). This is because the data disappear is MOL is no longer supported. If the data are available via the GBIF, the authors could share the script used to access them via the API. 

I would like to see the authors share the R script used in this study (e.g., via the OSF, Figshare, Zenodo, or some other trusted repository). Given that the aim of the study is to reduce information gaps, it would be nice to see the authors lead by example and readily share their data and code.

Minor comments:

L21. I suggest rephrasing this sentence as: "Here, we propose novel indicators of biodiversity data coverage and sampling effectiveness, and analyze national trajectories in closing spatiotemporal knowledge gaps for terrestrial vertebrates (1950-2019)."

L27. I suggest rephrasing this sentence as: "However, we found that a nation's coverage was stronger for species for which it holds greater stewardship."

L40. Missing reference

L45-47. Grammar

L64. It is unclear what you mean by "expectation". This only becomes evident later in the manuscript.

L67. Grammar. Key correlates of what? This sentence in unclear.

L71-72. This sentence is unclear… do you mean "allow changes in biodiversity data coverage over time to inform decision-making"?

L73. "nations […] stand to gain the greatest benefits from broadly improved biodiversity information" - why? It's unclear to me that this is the case for all nations.

L76-77. Can you cite one or multiple sources in support of this statement? What about the references cited on L229?

L79-81. This sentence is difficult to read. I suggest removing "an updatable framework and". Perhaps write a second sentence after this first one explaining what you mean by 'updatable framework'.

L88-90. I suggest modifying this sentence as follows: "We provide a first global assessment of… [expand] … for terrestrial vertebrates as well as infrastructure to continue tracking these indices at Map of Life (https://mol.org/indicators/coverage)."

L95. Grammar

L93-97. I suggest directing the reader to the relevant panel(s) of Fig. 1 after each term is explained. The current explanation of Steward's SSI is fairly limited in my opinion. If you cannot expand on it in the main text due to the word limit, I would direct the reader to the methods by adding "(see Methods") here rather than in the following sentence.

L102. Is the 'ideal uniform distribution based on Shannon's entropy' really the optimal sampling strategy? Again, I'm no expert in this field but I would have liked to see the rational for this choice, perhaps in the methods.

L110-112. This sentence in unclear.

L108-119. Please refer the reader to specific panels in Fig. 1 throughout this paragraph.

L121. I suggest using a word other than 'exploded'. 

L134. This sentence should be supported by a reference to a figure or paper.

L 143. Ref needed.

L144-147. The rational for this statement needs to be better explained.

L149-150. This sentence is unclear.

L155. Unclear.

L163. What is meant by "for the same number of records". Can you point the reader to a figure?

L170. Ref to figure needed.

L175. For clarity, please expand (perhaps in parenthesis) on what you mean by "effective and complementary".

L180. Given the importance of MOL in the context of this study, it would be helpful to have a box clearly explaining what it is.

L185-188. This is repeated twice, on L690-694 and L754-758. I have a really hard time making sense of Fig 5a based on this description and the figure caption…

L232-233. Grammar

L234. What are "limited transparencies of extrapolation approaches"? This sentence would benefit from being shortened. Perhaps split it into two?

I hope my comments are helpful.

Best regards,

Dom Roche

Reviewer #4:

[identifies himself as Jonathan Lenoir]

General comments

I read the work from Ruth Y. Oliver et al. with great interest and I think the authors are tackling an important topic which echoes the recent literature highlighting gaps and biases in our knowledge of global biodiversity distribution (and redistribution). In this manuscript, the authors focused on terrestrial vertebrates to assess both the global and national trends in the sampling effort to collect reliable data on species distribution. By doing so, the authors aim at highlighting, for each country separately, spatiotemporal trajectories in closing knowledge gaps in the distribution of terrestrial vertebrates. To achieve this goal, the authors built two species-specific metrics, namely (i) the species status information index (SSII) and (ii) the species sampling effectiveness index (SSEI), while accounting for country-specific stewardship of species (i.e. how responsible a country is for a given species, which is determined by the portion of the focal species global range that is occurring in the focal country) when aggregating across species the information per country. Using these two metrics, the authors found a rapid global increase, during the last decades, in the relative sampling coverage of each species distribution (SSII), especially so for some taxonomic groups, like birds. However, this rapid global increase in SSII further amplified the strong and existing geographic bias among countries, leading to country-specific SSII trajectories. Noteworthy, the authors suggested that the tremendous growth of species occurrence records in some countries failed to directly translate into newfound knowledge due to a sharp decline in sampling effectiveness (SSEI).

The manuscript is very well written and the figures are really helpful and informative, not only for displaying the main findings (Figs. 3-5) but also for helping the reader to understand the metrics used by the authors (especially Figs. 1 and 2). I really liked it as these figures are very intuitive, thus I would like to congratulate the authors for this effort. This said, I do have several important concerns and reservations that, I think, warrant publication of the manuscript as is. My first and main concern is about the SSEI metric which I think is somewhat flawed and inherently related to SSII. Indeed, the SSEI metric works such that the more occurrence records, the more likely it is that the distribution is uneven among the sampled grid cells. Let's for instance take the most extreme case of one single occurrence record for a given species. This means that only one grid cell among all the possible grid cells that are expected to be occupied is actually occupied and thus SSII is very low for that species, either globally or at the national level (except if the focal country is of the size of the occupied grid cell). For the SSEI metric, then the value is 1 just because the distribution is completely even across the occupied cells (1 occurrence record occurring in one grid cell: perfectly even). Hence, there is a mathematical relationship between SSEI and SSII, which are not fully independent from each other, such that the correlation can only be negative and the data constrained within an upper triangle when relating SSII (x-axis) against SSEI (y-axis). This is very well illustrated by the inset plot in Fig. 4d. As it is now, in my opinion, the SSEI metric is a bit useless, except towards very large values of SSII (when reaching completeness in the species distribution), because only then the SSEI metric bring new insights. However, for low SSII values, the SSEI has no meaning. A more meaningful and useful, I think, version of the SSEI metric should account for the total number of empty cells to further penalize the SSEI metric when only a few cells are occupied among the number of expected cells supposed to be occupied (cf. low SSII values).

My second major concern is about the very strong assumption that terrestrial vertebrate species have completely static distribution throughout the 70 years of the study period (1950-2019) and that any species range shifts would have little impact on the authors' analyses (see lines 570-572). This is especially problematic for the last three decades (1990-2019), during which evidence of species range shifts as climate warms were exploding (see Lenoir et al. 2020). Hence, I suggest the authors to carefully consider that important and recognized fact (species redistribution as global climate warms) in the scientific literature (Parmesan et al. 2003; Chen et al. 2011; Lenoir& Svenning 2015; Pecl et al. 2017; Lenoir et al. 2020) and at the very minimum discuss the implication of trans-boundary shifts in species distribution (e.g. species range expansions into new countries), which should alter the authors finding. Even better, but this involves quite some work I have to admit, would be to account for species range shifts by adjusting each species range map by means of species distribution models (SDMs), for instance. Alternatively, one cheaper solution involving less work for the authors would be to split the analyses into two periods to distinguish a period during which the authors' assumption is likely to hold (cf. a 30-yr period prior to climate warming: 1950-1979) and a second 30-yr period (1990-2019) during which species range shifts may alter the authors' findings, thus requiring to at least discuss the potential implications of trans-boundary shifts during this second time period. As for the authors' defence, terrestrial vertebrates are among the taxonomic groups showing the slowest (most often non-significant but not always: e.g. significant latitudinal range shifts for reptiles) velocities in species latitudinal range shifts, as opposed to marine species (see Fig. 3a in Lenoir et al. 2020). This said, terrestrial vertebrate species showed significant upslope range shifts (especially amphibians, but also birds and mammals: see Fig. 3b in Lenoir et al. 2020) during the last three decades. But I assume elevational range shifts will have relatively more minor impacts on trans-boundary shifts, except maybe for some highly mountainous countries like Bhutan, Nepal, Lesotho, Andorra, Chile or Switzerland where new species may arrive from the neighbouring lowland countries. Hence, I really urge the authors to at least discuss those important implications of trans-boundary shifts if they do not deem necessary (or if they think it is impossible) to account for species range shifts in their analyses, especially during the most recent (1990-2019) period.

Finally, as a relatively more minor concern, I think the authors should also better acknowledge the recent scientific literature highlighting the strong geographic, taxonomic but also methodological biases in species distribution and redistribution (as climate warms) which altogether suggests that no global biodiversity dataset or meta-analysis on biodiversity changes so far is truly global as it is most often claimed (see Brown et al. 2016; Feeley et al. 2017; Lenoir et al. 2020; Nunez & Amano 2021). See my specific comments below for more detailed suggestions which I hope the authors will find useful for their work.

Specific comments to the authors

Line 1: In the title, I think it is important to specify that this work focuses on terrestrial vertebrates only. Indeed, and although the same approach could be applied to other taxonomic groups as you mentioned in the text, this approach strongly relies on expert knowledge for species range maps, which is a strong limitation for a lot of very important taxonomic groups like plants or insects.

Line 26: The sharp decline in sampling effectiveness is mathematically expected as SSII increases (see my general comments).

Lines 34-35: Here, when mentioning the manifold consequences of biodiversity changes, you could cite some references like Pecl et al. (2017) or Bonebrake et al. (2018).

Lines 43-44: Indeed and databases on species range shifts already exist, like the BioShifts database (see Lenoir et al. 2020) which is freely available (https://doi.org/10.6084/m9.figshare.7413365.v1). 

Line 50: What about the need for long-term time series of monitoring data in order to assess biodiversity changes (see the BioTiME database from Dornelas et al. 2018)?

Lines 60-64: About taxonomic and geographic biases, the exact same pattern applies for data on species redistribution (see Feeley et al. 2017; Lenoir & Svenning 2015; Lenoir et al. 2020). Besides these biases, there is also important methodological biases in the way data are recorded and then used in subsequent quantitative analyses (see Brown et al. 2016; Lenoir et al. 2020). I think it is also worth pointing at in the list of biases that are currently acknowledged in the scientific literature.

Line 65: Not only socio-economic and ecological drivers to explain data gaps but also linguistic drivers. Indeed languages have also been highlighted as important barriers for global science in general (see Amano et al. 2016).

Line 75: About growing data for plant species distribution at the global extent, are you aware of the sPlot database (Bruelheide et al. 2019)? This one also suffers from the same geographic biases.

Line 85: See my general comment about the SSEI metric.

Line 92: About the optimal spatial resolution of your grid (110 km * 110 km at the equator), how did you set it exactly? Why do you consider this spatial resolution the most appropriate and finest spatial resolution you can use? Why not trying a finer spatial resolution? Did you run a sensitivity analysis varying the size of the spatial resolution? This was not clear in the methods section (cf. lines 569-570).

Lines 97-98: What about trans-boundary shifts under contemporary climate change? Some species are shifting poleward in latitude and upward in elevation, thus changing and potentially reshuffling the national stewardships you used in your analysis. This is an important matter (see my general comments), no?

Lines 101-102: Why constraining the SSEI metric to the realized spatial distribution of records? This is a bit misleading as you do not account for absence data in the other grid cells that are expected to be occupied. Yet, it is what matters in the end to get a uniform spatial distribution of records across all grid cells that are expected to be occupied, right? Why not penalizing the SSEI metric by the total number of grids cells supposed to be occupied but empty? Is it because you suppose that a grid cell expected to be occupied but empty could mean that the species is truly absent? To distinguish between that case (a true absence) and the other case of a false absence due to less sampling effort, you could use information from other species occurrence records during the same time period. I mean, if there is a high density of occurrence records for other terrestrial vertebrates in a grid cell where the focal species is absent but supposed to be occurring, then it is likely to be a true absence, right?

Lines 112-113: Indeed, but this is expected since the more occurrence records you sample, the more likely it is to be unevenly distributed throughout a relatively larger number of occupied grid cells (higher SSII values).

Lines 121: Indeed, see also the recent release of: sPlot, the global vegetation plot database (Bruelheide et al. 2019) providing data on plant co-occurrence; BioTIME, the global database on biodiversity time series (Dornelas et al. 2018); or BioShifts, a database on species range shifts (Lenoir et al. 2020).

Lines 128-129: Again, this is very much expected and not surprising given how SSEI works (see my general comments).

Line 135: What do you mean exactly by "slowed in their coverage progress"? Do you mean according to SSII or according to SSEI? Or both?

Lines 150-153: According to the realized distribution yes, it is true that sampling effectiveness has decreased in countries with increased SSII (expected pattern), but according to the expected distribution, this is not necessarily the case, right? Again, I think the way the SSEI is constructed is problematic as it is specifically focused on the realized distribution of records and it is completely blind to the expected distribution of records (cf. it does not account for the distribution of cells expected to be occupied but empty).

Lines 165-170: This pattern for birds (increasing SSII but decreasing SSEI) is likely due to the fact that citizen science data are everything except strategic sampling and thus the increase in spatial coverage in occurrence records automatically comes with a decrease in the spatial evenness of those records across the sampled area, this is well expected (see my general comments). The more data we collect, the more it is likely to be unevenly distributed in space because humans never collect data randomly in the field. Once a good spatial coverage in occurrence records is achieved, one needs to invest in strategic sampling, which cannot really be achieved by opportunistic data, except under strong guidance by the scientific community to orient citizen science data towards a strategic sampling. I think this is something important to discuss. The rapid increase in citizen science data in ecology is a great thing but it is definitely not the most efficient approach to reach a strategic sampling design if citizen science is not undertaken under the guidance of the scientific community. Not even mentioning the issue of data quality, this type of opportunistic and chaotic data (in terms of spatial distribution) comes with costs as it is then very difficult to use and analyse such data when monitoring biodiversity changes over time. Opportunistic data as collected by GBIF cannot really replace a strategic sampling that is designed for the purpose to assess biodiversity changes. Maybe this could be discussed and highlighted to better balance the discussion around citizen science data.

Lines 172-174: Ok, but this comes with costs (see my comment just above) and it should be discussed. I have the impression that the discussion is only oriented towards the benefits of citizen science data without discussing its drawbacks (e.g. uneven distribution of data).

Lines 176-178: Indeed, this is very important and you could expand a bit this part of the discussion (see my comments above).

Lines 183-184: Linguistic factors as well (Amano et al. 2016).

Lines 184-188: Why did you only use the SSII metric when building your categories? I find it a bit strange that you did not also incorporate the SSEI metric in your categorization. Maybe this is somewhat related to my comment on the fact that SSEI is a bit meaningless under low values of SSII.

Lines 210-212: Can you provide examples of countries reflecting this situation?

Lines 214-225: Here, you discuss the drawbacks of the SSII metric, which is nice, but what about the drawbacks of the SSEI metric (cf. my general comments on the issue that SSEI is not really informative for low values of SSII)? Besides, when discussing the issue of invasive species for the SSII metric, you should also remind the reader about the static nature of SSII (cf. you do not account for potential species range shift under climate change) and the fact that trans-boundary shifts of native species under anthropogenic climate change may completely change the pattern you found for SSII, and especially so for steward's SSII.

Line 235: Species-level expectations should account for species redistribution under contemporary climate change, this is not a minor issue (see my general comments).

Line 238: Please provide citations to illustrate this increase in mapping efforts for other taxonomic groups and systems such as plants (see Bruelheide et al. 2019) and marine taxa (cf. OBIS data).

Lines 322-323: Shading are not visible in panel c. Is it because of very low 95%CI?

I sincerely hope that my comments and suggestions will help both the authors to improve their work as well as the editorial board to take the right decision on the present manuscript. It was a real pleasure to read and review this inspiring work.

Jonathan Lenoir

References

Amano et al. (2016) Languages Are Still a Major Barrier to Global Science. PLoS Biology, 2000933

Bonebrake et al. (2018) Managing consequences of climate‐driven species redistribution requires integration of ecology, conservation and social science. Biological Reviews, 93, 284-305

Brown et al. (2016) Ecological and methodological drivers of species' distribution and phenology responses to climate change. Global Change Biology, 22, 1548-1560

Bruelheide et al. (2019) sPlot - A new tool for global vegetation analyses. Journal of Vegetation Science, 30, 161-186

Chen et al. (2011) Rapid range shifts of species associated with high levels of climate warming. Science, 333, 1024-1026

Dornelas et al. (2018) BioTIME: A database of biodiversity time series for the Anthropocene. Global Ecology and Biogeography, 27, 760-786

Feeley et al. (2017) Most "global" reviews of species' responses to climate change are not truly global. Diversity & Distribution, 23, 231-234

Lenoir & Svenning (2015) Climate-related range shifts - a global multidimensional synthesis and new research directions. Ecography, 38, 15-28

Lenoir et al. (2020) Species better track climate warming in the oceans than on land. Nature Ecology & Evolution, 4, 1044-1059

Parmesan et al. (2003) A globally coherent fingerprint of climate change impacts across natural systems. Nature, 421, 37-42

Pecl et al. (2017) Biodiversity redistribution under climate change: Impacts on ecosystems and human well-being. Science, 355, eaai9214

---

## [Decision Letter · Decision Letter 2]

18 May 2021

Dear Dr Oliver,

Thank you for submitting your revised Research Article entitled "Global and national trends in documenting and monitoring species distributions" for publication in PLOS Biology. I have now obtained advice from three of the original reviewers and have discussed their comments with the Academic Editor. 

Based on the reviews, we will probably accept this manuscript for publication, provided you satisfactorily address the remaining points raised by the reviewers. Please also make sure to address the following data and other policy-related requests.

IMPORTANT:

a) Your findings are somewhat nuanced, but we wondering whether they could be indicated in some way in your Title to make it more informative (something like "Global and national trends in documenting and monitoring species distributions identify opportunities to close critical information gaps").

b) Please attend to the remaining requests from revs #3 and #4. The Academic Editor particularly wanted me to stress the need to ensure that the precedence for SSEI is made clear, as requested by rev #4 ("I would like to see the authors acknowledge the equivalence of SSEI with preceding indices (Lenoir review) and tone down claims of novelty. The results are important without this claim.").

c) You'll see that rev #1 raises strong concerns about data availability (as did rev #3 in the previous round). This is clearly an important and emotive issue in the community, and we encourage you to do your utmost on this front. Specifically, we remind you of your commitment previously expressed in the Response to Reviewers: "MOL has submitted an application to be formally recognized as a trusted data repository. If that status is not in place by the time of publication, all information necessary to replicate the results will in addition to on MOL also be made available on a formally already recognized

repository."

d) You should also familiarise yourself with our data policy. This does have exemptions for third party data, but please could you provide information on the license or a letter/email from the third party (IUCN), explicitly stating that the data cannot be shared even with appropriate credits? If the data can be shared, then you should do so.

e) Please attend to my Data Policy requests below. Specifically, we will require any code needed to reproduce your results to be made available as supplementary files or in e.g. Github. We'll also need the numerical values underlying Figs 2CDEFG, 3ABCDEF, 4CF, 5ABC, S1ABCD, S2AB, S3CDEF. In addition, please cite the location of the data clearly in each relevant Fig legend.

We expect to receive your revised manuscript within two weeks. 

*Published Peer Review History*

*Early Version*

Sincerely,

Roli Roberts

Senior Editor,

rroberts@plos.org,

PLOS Biology

DATA POLICY:

Please make all code required to reproduce your results available, either as a supplementary file or in a repository such as Github. We also require all the numerical values underlying the figures and results of your paper be made available in one of the following forms:

Regardless of the method selected, please ensure that you provide the individual numerical values that underlie the summary data displayed in the following figure panels as they are essential for readers to assess your analysis and to reproduce it: Figs 2CDEFG, 3ABCDEF, 4CF, 5ABC, S1ABCD, S2AB, S3CDEF. NOTE: the numerical data provided should include all replicates AND the way in which the plotted mean and errors were derived (it should not present only the mean/average values).

DATA NOT SHOWN?

REVIEWERS' COMMENTS:

Reviewer #1:

I sincerely appreciate the work put forth by the authors here. After seeing the decision come through, I thought "wow - four massive reviews - that is a lot of work". I'm glad, however, that the authors decided to undertake this work. I believe they struck a nice balance of appeasing Reviewers and also strengthening their manuscript, without going down too many rabbit holes. Well done.

All in all, I believe the text is sufficiently revised for publication.

BUT, I have one sticking point. And ultimately, this is up to the Editorial Board at PLoS Biology. This was also highlighted by Reviewer 3 in the original round of review. That is 'data availability'. 

I mean no offense, but I strongly disagree with the notion of "visually accessible" (the wording in the current data availability statement). To be abrupt, I don't think I've seen this wording yet. And I honestly don't know what this means, but to me, it does not meet the notion of data accessibility/availability that we as a conservation biology community should be striving for. And even for the indicators information, I still don't know how to get these data. I went to https://mol.org/indicators/ as suggested by the data availability statement. But I still don't see any 'download' button or any way to get the data. All I see is 'explore'. I believe that an 'account' should not need to be created in order to access these data (I'm partially assuming that is how I can download the entire dataset). These data need to be made available with the paper as a current record of downloadable and reproducible research in my opinion. All the data, not simply one country at a time. Especially, for such a prestigious outlet of PLoS Biology. I understand the idea of IUCN not making range data available (which is a sticky problem in and of itself) and your hands being tied, so to speak. But regardless, the authors should strive to make as much data available as possible to reproduce these results. I apologize if this sounds harsh, but I think if real change is going to ever happen in our (the scientific community, specifically ecology/biodiversity research) push for available data and reproducibility then it starts with big/prestigious labs (e.g., Walter's group) publishing papers in big outlets (e.g., PLoS Biology) going the extra mile to ensure the data are available and results reproducible beyond "visually accessible".

Best of luck with your future work and I look forward to seeing this manuscript online soon!

Reviewer #3:

[identifies himself as Dominique Roche]

The authors have put considerable work into addressing the reviewer comments and . The only comment I have pertains to the need to present test statistics (and effect sizes, when possible) alongside p values in the main text and the ESM. P values should also be reported to three decimal places. Although not necessary, it would be helpful if the authors also included the sample size or degrees of freedom as appropriate. I cannot indicate line numbers as there are none in the manuscript. Examples include:

Main text:

"Globally, only approximately half of nations (42%) showed increasing, significant trends (p < 0.01) in coverage averaged across taxa over the previous decade (Fig. 4b)."

Captions for Figs 4 and 5. The term 'significant' is used repeatedly without reference to the ESM where the methods and statistical results are presented.

ESM:

"The temporal patterns in SSII are different, with birds only exceeding the three other groups

after 1980, but since then showing near linear-growth in taxon-wide SSII and exceeding other

classes in 2019 by nearly 10-fold (Fig. 3c)." What statistics were used to infer these results? If it was a comparison of confidence intervals, perhaps re-state here for clarity.

"Steward's SSII has recently increased in a majority of nations (84%), particularly in North

America and southern and eastern Europe with nearly half of nations (42%) showing significant

(p < 0.001) increasing trends (Fig. 3b). Of the minority (13%) with decreasing rates, Finland had

the most rapid decrease (-0.021 SSII/year). Despite mostly positive trends, much of Africa and

Asia saw only negligible increases in indicator values over the last decade, with the exceptions

of India, Sri Lanka, and South Korea which showed large increases in data coverage. Nations

were nearly evenly split between either non-significant and significantly increasing Steward's

SSII for resident bird species (52.8% and 47.2%, respectively, none decreasing; Fig. 3c). Most

nations did not have significant trends in data coverage for mammals (85.8%), amphibians

(89.9%), and reptiles (81%)."

"Recent National SSEI differed strongly among nations (Fig. 4d, Supplementary Table 3).

National SSEI was generally lower within western Europe, North America, and Australia.

National SSEI and Steward¶s SSII were weakl\\, negativel\\ correlated (Spearman¶s rho = -0.52,

p < 0.001). A majority of nations (51%) had decreasing SSEI across terrestrial vertebrates,

however only 11% of nations globally had significant (p < 0.01), decreasing trends (Fig. 3e).

These nations included the United States, Canada, Italy, and South Africa. Decreasing trends in

SSEI were most common for bird species (27.5%) (Fig. 4f)."

etc.

Reviewer #4:

[identifies himself as Jonathan Lenoir]

General comments

I was one of the four reviewers (reviewer #4) during the former round of review. I read the authors' responses to my comments (as well as their responses to the comments from the other three reviewers) and I particularly appreciate the efforts made by the authors to address most of the concerns collectively arising from the four reviewers. For instance, the new Supplementary Fig. 3 on the impact of varying grain sizes on the SSII metric is a great addition in terms of sensitivity analysis, as requested by reviewers #1 and #3. But then, why not also assessing the impact of varying spatial grains on the other metric: SSEI? Or is it because the SSEI metric is unaffected by the variation in the spatial grain? If so, it would be nice to precise it and show it.

About SSEI, I am also very grateful to the authors for considering my main initial concern on the potential link between SSII and SSEI. The additional explanations provided by the authors and the new panel in Supplementary Fig. 4 are really helpful in that respect. Also, the important clarification on the special case of just a single grid cell where there is one or several records of occurrence for the focal species is indeed important. In fact, such cases would lead to a maximum entropy (H*) being equal to zero (log(1)=0) and thus the SSEI index would equal Inf value. So, I am grateful to the authors for also clarifying this issue.

This said, I am still convinced that the SSEI metric is only useful when N, the total number of records (i.e. total number of occurrence records for a given species), is several order of magnitude larger (e.g. 10 to 100 times at least) than G, the total number of grid cells where there is actual information on sampling effort. Indeed, both panels (a) and (b) in Supplementary Fig. 4 are actually illustrating this very well since SSEI can only reach very low values and thus be highly variable (which is a very important feature for discriminating different sampling effectiveness situations) when the total number of records is way larger than the total number of grid cells sampled. So, I would at least recommend to discuss that inherent property of the SSEI metric and recommend the authors to warn the reader about this and that the SSEI metric will be especially relevant under high sampling effort (in terms of total number of occurrence records) relative to the number of grid cells that are sampled.

Still about the SSEI metric and information theory (IT), I would like to mention here that this metric is, simply put, the Pielou's index of diversity, also called equitability or evenness index (cf. the empirical entropy measured by Shannon's index divided by the maximum entropy given by Shannon's index). The analogy with the way the Shannon's index is used in Ecology to measure species richness is that maximum entropy is given by H*=log(S) where S is the total number of species and then N is the total number of individuals, with ni being the number of individuals for species i. Here, the authors did not consider S but G, the total number of grid cells sampled with N being the total number of occurrence records and not the total number of individuals. Hence, there is nothing new or novel for me with the SSEI index itself because it is a metric that already exists in IT and that is widely used in ecology. Hence, the authors' claim in the abstract that "we propose novel indicators of biodiversity data coverage and sampling effectiveness" is an overstatement to me. Yet, the idea of the authors to borrow the Shannon's and Pielou's index from IT for assessing the equitability of the sampling of occurrence records over the grid cells where the species is known to occur is indeed a novel application of the Pielou's index and an interesting one. This is why I think this study has definitely a great potential, not because of the SSEI metric itself (nothing new with that because it is Pielou's index) but because of its application to assess sampling effectiveness across the set of grid cells where data is available. So, maybe it would be nice that the authors tone down this claim of "novelty" on the SSEI metric and actually refer to the Pielou's index when mentioning the SSEI metric, just to relate to existing metrics from IT. By the way, I actually think there is a mistake in the text explaining the SSEI metric as the formula written in the Supplementary Materials, Methods section (SSEI subsection), for H* (cf. H*=log(N)) is wrong. Indeed, N is the total number of records here, while the formula of maximum entropy should be H*=log(G), where G is the total number of sampled grid cells. I assume that only the formula of H* in the text of the Supplementary Materials is wrong and that the authors actually used the right formula in their computations and analyses, but it would be nice that the authors confirm this is actually the case. Note that the authors are sometimes mixing G with N in their responses to my former comments. For instance, when mentioning the special case of a single grid cell (G), the authors mentioned a single occurrence record (N), which is different. Indeed, the same issue applies with several occurrence records if they all fall inside the same grid cell Gi, so it is the total number of sampled grid cell G that matters here and which should be strictly greater than 1.

Sorry to insist on the metrics used here but these are quite central to the whole study. In that respect, I invite the authors to also consider the Simpson's index for assessing the evenness and equitability of the sampling across the sampled grid cells (cf. SSEI). Indeed, the Simpson's index has the advantage to account for the total number of grid cells that are actually sampled which is not the case for the Pielou's index used by the authors. Indeed, under perfectly even sampling effort, Pielou's index will give the exact same value of 1, or perfect evenness, if just 2 or 1000 grid cells are sampled while the Simpson's index will give a higher value for the situation in which more grid cells are sampled.

About my second major concern regarding species range shifts over time, the authors did a good job to address this point and to acknowledge the existing scientific literature on the matter. I have nothing else to add at this stage regarding this second concern I had. I agree with the authors that it would be too much to integrate in this study the temporal dynamic of species range shifts but it is good that the authors discuss their approach in light of this and that it can be implemented in the future given the increasing amount of biodiversity time series.

Again, I would like to thank the authors for addressing and answering my initial comments and concerns. I hope these new suggestions in light of the revised version of their work will further help them.

Yours sincerely,

Jonathan Lenoir

---

## [Editor Report · Decision Letter 3]

22 Jun 2021

Dear Ruth,

On behalf of my colleagues and the Academic Editor, Craig Moritz, I'm pleased to say that we can in principle offer to publish your Research Article "Global and national trends, gaps, and opportunities in documenting and monitoring species distributions" in PLOS Biology, provided you address any remaining formatting and reporting issues. These will be detailed in an email that will follow this letter and that you will usually receive within 2-3 business days, during which time no action is required from you. Please note that we will not be able to formally accept your manuscript and schedule it for publication until you have made the required changes.

IMPORTANT: Many thanks for clarifying the data provision. Please could you also include clear mentions of its location in each relevant main and supplementary Figure legend? e.g. "The data underlying this Figure may be found in https://github.com/MapofLife/biodiversity-data-gaps". This may look repetitive, but we want each Figure (and its data) to be standalone. I've flagged to my colleagues that I've requested this change.

PRESS: We frequently collaborate with press offices. If your institution or institutions have a press office, please notify them about your upcoming paper at this point, to enable them to help maximise its impact. If the press office is planning to promote your findings, we would be grateful if they could coordinate with biologypress@plos.org. If you have not yet opted out of the early version process, we ask that you notify us immediately of any press plans so that we may do so on your behalf.

Sincerely, 

Roli

Roland G Roberts, PhD 

Senior Editor 

PLOS Biology

rroberts@plos.org